# Visible Light Assisted Organosilane Assembly on Mesoporous Silicon Films and Particles

**DOI:** 10.3390/ma12010131

**Published:** 2019-01-03

**Authors:** Chloé Rodriguez, Alvaro Muñoz Noval, Vicente Torres-Costa, Giacomo Ceccone, Miguel Manso Silván

**Affiliations:** 1Departamento de Física Aplicada and Instituto de Ciencia de Materiales Nicolás Cabrera, Universidad Autónoma de Madrid, 28049 Madrid, Spain; chloe.rodriguez@uam.es (C.R.); alvaro.betelgeuse@gmail.com (A.M.N.); vicente.torres@uam.es (V.T.-C.); 2Centro de Microanálisis de Materiales, Universidad Autónoma de Madrid, 28049 Madrid, Spain; 3European Commission, Joint Research Center, 21020 Ispra (Va), Italy; giacomo.ceccone@ec.europa.eu

**Keywords:** porous silicon, visible light assisted organosilanization, solid state NMR, XPS, ToF-SIMS

## Abstract

Porous silicon (PSi) is a versatile matrix with tailorable surface reactivity, which allows the processing of a range of multifunctional films and particles. The biomedical applications of PSi often require a surface capping with organic functionalities. This work shows that visible light can be used to catalyze the assembly of organosilanes on the PSi, as demonstrated with two organosilanes: aminopropyl-triethoxy-silane and perfluorodecyl-triethoxy-silane. We studied the process related to PSi films (PSiFs), which were characterized by X-ray photoelectron spectroscopy (XPS), time of flight secondary ion mass spectroscopy (ToF-SIMS) and field emission scanning electron microscopy (FESEM) before and after a plasma patterning process. The analyses confirmed the surface oxidation and the anchorage of the organosilane backbone. We further highlighted the surface analytical potential of ^13^C, ^19^F and ^29^Si solid-state NMR (SS-NMR) as compared to Fourier transformed infrared spectroscopy (FTIR) in the characterization of functionalized PSi particles (PSiPs). The reduced invasiveness of the organosilanization regarding the PSiPs morphology was confirmed using transmission electron microscopy (TEM) and FESEM. Relevantly, the results obtained on PSiPs complemented those obtained on PSiFs. SS-NMR suggests a number of siloxane bonds between the organosilane and the PSiPs, which does not reach levels of maximum heterogeneous condensation, while ToF-SIMS suggested a certain degree of organosilane polymerization. Additionally, differences among the carbons in the organic (non-hydrolyzable) functionalizing groups are identified, especially in the case of the perfluorodecyl group. The spectroscopic characterization was used to propose a mechanism for the visible light activation of the organosilane assembly, which is based on the initial photoactivated oxidation of the PSi matrix.

## 1. Introduction

The functionalization of semiconductor nanostructures with organic monolayers is a requirement for tailoring their surface chemistry for ulterior bioconjugation [1,2]. Porous Silicon (PSi) can be described as a matrix of silicon quantum dots (Si QDs) immersed in an amorphous Si/silica network. It differs from Si QDs in amorphous Si networks in the high specific surface areas (hundreds of m^2^/g) [3], tailorable within a wide range of pore structures. Although both micro- and mesopores can be used to achieve high porosity, mesoporous structures are generally preferred because they exhibit better mechanical stability in comparison to the microporous ones [4]. With respect to porous silica networks, they differ mainly in their optoelectronic properties (photo and electroluminescence). PSi is generally engineered as a film supported on a Si wafer (PSiF), but has been described also in the form of colloidal particles (PSiPs) for a wide range of applications.

Engineered PSi is especially attractive in the biomedical field due to its high biocompatibility, biodegradability [5] and role as multifunctional actuator [6]. By means of functionalization, PSi had gained a great deal of attention in applications such as magnetic focusing [7] or drug delivery [8], targeted contrast agents in biomedical imaging [9], tissue engineering through polymer [10] or hydroxyapatite loads [11], photothermal cancer treatment through plasmonic composites [12], biocatalytic substrate upon functionalization with enzymes [13] or biosensing through interferometric effects [14]. In order to avoid an unspecific reactivity of PSi, a conjugation with organic molecules/biomolecules is essential.

Surface modification of PSi through organosilane assemblies allows for specific functionalities and biomolecular selectivity [15]. Relevantly, to enable the organosilane assembly, a prior oxidation of the PSi surface is required, which has been traditionally carried out by thermal or chemical oxidation [16]. Such oxidation of PSi has previously been induced by visible light activation, but aimed at a functionalization with metal nanoparticles. Light assisted redox reactions with residual water are in fact responsible for a mild surface oxidation [17], which we have previously proposed for the activation of the assembly of glycydyloxy-trimethyl-silane [18].

In this work, we aim to propose a generalization for the process of visible light assisted organosilanization (VLAO) with two additional organosilanes, as well as a reaction mechanism for the process. We therefore rely on an advanced characterization of the modified surfaces. We outline the analytical potential of solid-state nuclear magnetic resonance (SS-NMR) at the surface level and complement with traditional surface spectroscopic techniques, such as time of flight secondary ion mass spectroscopy (ToF-SIMS), and X-ray photoelectron spectroscopy (XPS), in order to study the properties of organosilane functionalized PSi. The surface sensitivity of SS-NMR has already been highlighted in porous silica based systems as a technique to decipher the influence of surfactants in the structure of molecular sieves [19]. Within the more restricted area of PSi, SS-NMR has been used for the characterization of the chemistry upon formation [20] and the correlation of structure with luminescence [21].

Contrary to bulk Si, where the surface is inaccessible to NMR, PSi provides a sufficient number of surface nuclei to overcome NMR’s inherently low surface sensitivity and achieve a satisfactory signal-to-noise ratio [22]. Relevantly, although the sensitivity of SS-NMR for amine surface capping has already been outlined for Si nanoparticles [23], a prospective study on the analytical potential of SS-NMR on organosilane functionalized PSiPs could provide further insight into the chemical structure of the formed hybrid materials. A surface/interface sensitive approach of this sort has already been used to characterize the organosilane adsorption on highly porous clay materials [24]. Transmission electron microscopy (TEM), scanning electron microscopy (SEM), Fourier transformed infrared spectroscopy (FTIR), ToF-SIMS and XPS are used on convenient PSiF and PSiP models to characterize the organosilane assemblies and correlate with the SS-NMR information.

## 2. Experimental

### 2.1. Preparation of PSiFs and PSiPs

The back side of the p-type boron-doped (resistivity 0.05–0.1 Ω·cm) (100) oriented Si wafers was first coated with an aluminum layer to provide low resistance ohmic electrical contacts. Si was then cut and mounted into a Teflon electrochemical cell to form PSiFs by anodic etching of the silicon wafers in an aqueous electrolyte composed of a mixture of hydrofluoric acid (HF) (40%) and absolute ethanol (98%) (volume ratio 1:1). The current density was fixed at 78 mA/cm^2^ and the anodization time at 20 s, leading to a 1 µm thick PSiF. For the formation of PSiPs, the current was periodically (every 50 s) pulsed for 1 s to a higher value (104 mA/cm^2^). The waveform was repeated for 30 cycles, producing highly porous and mechanically fragile layers spaced at predetermined points in the porous film [25]. The thickness-induced instability in the PS/Si interface caused a fragmentation of the layer, which could be easily scrapped from the surface. Through these artificial cleavage planes in the PSi film, we favored the extraction of homogeneous PSiPs by sonication in ethanol for 3 h. The resulting PSiPs were rinsed with ethanol and used without any filtration.

### 2.2. Organosilane Assembly on PSi

Two different chemical functionalities were considered using 3-aminopropyl-triethoxy-silane (APTS) and 1*H*,1*H*,2*H*,2*H*-perfluorodecyl-triethoxy-silane (PFDS) (both from Sigma Aldrich, St. Louis, MO, USA) (Figure 1). The organosilane-based solutions were prepared by diluting the aminosilane (0.2% *v*/*v*) in EtOH (absolute, 99.8%, Sigma Aldrich). The VLAO process took place by illuminating the concerned PSi objects through the silane dilutions so that both the PSi structures and the organosilanes were exposed to the visible light simultaneously. The visible light source consisted of a 150 W halogen lamp with a maximum of 10% power emission at 700 nm and a power emission below 2% for radiation under 360 nm. The VLAO process on the PSiPs/PSiFs took place for 10 min (APTS) or 30 min (PFDS). Temperature rises (T < 40 °C) were limited with a heat dissipation bath [18]. The silane concentration used for the functionalization was low in order to remain within the sub-monolayer regime and minimize undesired reactions with solvent moisture. Residual water was however not extracted from the solvent due to its principal role in the photo-assisted oxidation of the PSi surface [26]. The samples were then cleaned by rinsing in the respective solvent used for the reaction and dried under an N_2_ flow. The whole process was carried out in a glove box.

The surface characterization of PSiFs was performed on samples bearing an internal negative control. After visible light activated organosilane assembly, PSiFs were patterned using a Si mask in an etching process carried out in a capacitive plasma reactor, for 5 min, with a 50 W RF (13.56 MHz) discharge, 30 sccm Ar and a working pressure of 65 Pa.

### 2.3. Characterization

Morphological characterization of the PSiPs was carried out using a 2100F TEM (JEOL, Akishima, Tokyo, Japan) operated at 200 kV after dispersing the PSiPs from an acetone solutions onto carbon coated Cu grids. Field emission SEM (FESEM) images from PSiPs and PSiFs were acquired in a XL30S microscope (FEI/Philips, Hillsboro, OR, United States) operated operated at 10 keV. The characterization of the functionalized PSiPs was performed using a Vector 22 FTIR spectrometer (Brucker, Billerica, MA, US), resolution 8 cm^−1^, 4000–400 cm^−1^, 32 scans at 10 kHz) in the transmission configuration after the preparation of KBr disks. NMR spectra were obtained at room temperature in a AV-400-WB (Bruker, Billerica, MA, US) incorporating a 4 mm triple probe channel using ZrO rotors with KeI-F stopper. The rotor speed was 10 kHz. ^13^C spectra were obtained in a cross-polarization (CP-MAS) between ^1^H and ^13^C nuclear spins with dipolar decoupling of ^1^H at 80 kHz. The working frequencies were 400.13 MHz for ^1^H and 100.61 MHz for ^13^C. The ^1^H excitation pulse was 2.75 µs, the spectral width was 35 kHz, the contact time was 3 ms, the relaxation time was 4 s, and 1 k scans were accumulated. TMS and Adamantano (CH_2_ 29.5 ppm) were the primary and secondary references, respectively. ^19^F spectra were acquired with a double probe channel of 2.5 mm and Vespel stopper rotors using 2 μs pulses of 90° and a spectral width of 30 kHz. The recovery time was 120 s due to ^19^F nuclei having large spin-lattice relaxation times [27]. The rotation speed was 20 kHz and the spectra were acquired overnight. CFCl_3_ and Na_2_SiF_6_ (−152.46 ppm) were used as the primary and secondary references, respectively. For the ^29^Si cross polarization magic angle spinning (CP/MAS) experiment, an excitation pulse of 3 μs was used for ^1^H, corresponding to an angle of π/2. The relaxation time was 5 s and the spectral width 40 kHz. TMS and Caolin (−91.2 ppm) were used as the primary and secondary references, respectively. The ‘magic angle’ of 54.74° relative to the direction of the static magnetic field allow the elimination of broadening.

XPS data were obtained with a with a PHOIBOS 150 9MCD energy analyzer (SPECS, Berlin, Germany). XPS spectra of the functionalized PSiFs were acquired with electron emission angles of 0° and source-to-analyzer angles of 90° using non-monochromatic Mg K_α_ excitation with pass energies of 75 eV for the survey and 25 eV for the core-level spectra, respectively. The data were processed using CasaXPS v16R1 (Casa Software, UK) taking the aliphatic contribution to the C1s core level at 285.0 eV as a binding energy (BE) reference. The PFDS micropatterned PSiFs were characterized with ToF-SIMS. The analysis was conducted using a reflector type SIMS IV spectrometer incorporated with Surface Lab software v6.4 (ION-TOF GmbH, Münster, Germany) and a 25 keV liquid metal ion gun (LMIG) operating with bismuth primary ions. Spectra were acquired in static mode (primary ion fluence <10^12^ ions·cm^−2^). Charge compensation was applied to the analysis using low-energy (~20 eV) electron flooding. Mass calibration was obtained using the peaks C- (12 *m*/*z*), C2- (24 *m*/*z*), C3- (36 *m*/*z*), C4- (48 *m*/*z*), and C5- (60 *m*/*z*) for negative ion and H+, C+ (12 *m*/*z*), CH+ (13 *m*/*z*), CH2+ (14 *m*/*z*) and CH3+ (15 *m*/*z*) for positive ion spectra. Analyses were done using a squared area of 250 × 250 μm^2^ in the high mass resolution burst mode (resolution M/ΔM > 6000). Values of *m*/*z* are given as dimensionless in keeping with international union of pure and applied chemistry (IUPAC) recommendations, even though *m* represents the unified atomic mass unit in *u*.

## 3. Results and Discussion

### 3.1. Characterization of PSiFs

The effects of the VLAO process on the PSiFs was initially evaluated using FESEM. The results illustrate that the process respects the initial topography of a PSiF control (Figure 2a), where the different magnifications of the process performed using both APTS (Figure 2b) and PFDS (Figure 2c) denote no pore occlusion or drastic adsorption of heterogeneously induced colloidal structures [28], which are often the by-product of silanization processes. Although the porosity remains within the same order of magnitude for both samples, the resulting mean pore size is slightly bigger for APTS functionalization (roughly 40 nm) as compared to the PFDS functionalization (approximately 25 nm). The origin of this difference resides in the different porosities observed on batch-to-batch observation due to slight differences in the radial distribution of the electric field on the active Si electrode and the loss of chemical potential of the electrolyte during the progression of one series of samples.

To further explore the properties of VLAO processed PSiFs, we performed a surface analysis of the samples functionalized with the two organosilanes before and after an Ar plasma etching process. The APTS functionalized PSiFs were characterized by XPS as shown in Figure 3, by acquiring spectra from macro-patterned areas (i.e., a 1 × 1 cm^2^ Si mask-exposed half of the APTS-PSiF to the plasma protecting the rest of the sample). The protected area (see spectra on the top of Figure 3) showed a considerably high content of C (22.0 at.%) and N (4.4 at.%), sustaining the integration of the aminopropyl group on the surface. O (41.2 at.%) and Si (32.4 at.%) however, were the dominant elements on the surface as shown in the top, widescan spectrum image in Figure 3. The analysis of the most relevant core level spectra confirmed on the one hand, the variety of the surface organic species, and on the second hand, the predominance of the oxidized state of the Si species. In fact, the Si2p core level spectrum (center) shows a dominant component with a BE at 103.0 eV, compatible with a fully oxidized Si, which can arise from both the oxidized PSiF [29] and the siloxane bonds from the adsorbed organosilane [30]. Relevantly, the C1s core level is characterized by a strong asymmetry, compatible with four different contributions [31,32]: The aliphatic C–C at BE 285.0 eV, the C–N at BE 286.0 eV, the C–O at BE 287. 2 eV and the C=O at BE 288.5 eV. The first two contributions of the C1s spectrum (right) confirm the high level of integration of the aminopropyl group, while the N1s spectrum in the inset shows that the primary amines are present in neutral and charged forms [33,34] for the low and high BE contributions, respectively.

After the selective Ar plasma etching process, the surface composition of the APTS functionalized PSiFs changed considerably as shown in the set of spectra in the bottom part of Figure 3. The etching treatment had a drastic effect on the organic content of the PSiFs with a reduction from 22 to 13 at.% in C and an almost complete deletion of surface N (only traces below 1% could be detected, see corresponding widescan spectrum). Further, the Si content was drastically affected with a reduction from 35 to 21 at.%.

The surface suffers a drastic C depletion after the etching process with a resulting increase of O content from approximately 41 up to 62 at.%. Additionally, the presence of F produced as a by-product during the PSiF synthesis was confirmed after etching of the adsorbed APTS (up to 3.3 at.%). The preferential effect on deletion of the adsorbed APTS was confirmed from the Si2p spectrum. After etching, a secondary wide minor contribution at the low BE side was observed, denoting the presence of Si–Si clusters near the surface, which are compatible with the structure of non-fully oxidized PSiFs [29]. With regard to the C1s spectrum, the relative intensity of the contributions at a high BE was drastically reduced, which denotes the drastic reduction in variety of organic structures on the PSiF surface.

The PSiFs functionalized with PFDS were characterized using ToF-SIMS, as illustrated in Figure 4 for two different regions (positive and negative on the left and right, respectively). During the Ar plasma etching process, a micro-mask was used to exploit the imaging mode of the equipment used. In the positive ions spectra, the mask protected areas denoted the presence of C_2_H_3_O^+^ and SiHO^+^ ions, which are characteristic derivative fragments of the core of perfluorinated silanes [35]. On the other hand, the plasma exposed areas showed contributions from KNaF^+^ and K_2_F^+^, which are characteristic of F by-products of the PSiF formation process [36]. The same kind of relationships were found in a second region used for the analysis of negative ions. Characteristic ions from masked protected areas, such as Si_3_HO_7_^−^ or Si_3_C_3_HO^−^, can be ascribed to a certain level of ‘clusterization’ of siloxane bonds from PFDS, suggesting that the VLAO process leads to some homogeneous condensation of PFDS. For the exposed areas, the SiF_5_^−^ and F^−^ ion maps illustrate once again, that the Ar plasma process induces an efficient surface cleaning, exposing by-products of the PSiF synthesis.

Overall it can be concluded that APTS- and PFDS-rich PSiFs can be formed by a VLAO process and that the activation of a certain degree of poly-silane bonds on the surface is strongly suggested. A selective Ar plasma etching process can reverse the presence of the organosilanes locally, re-exposing the underlying PSiF to the surface.

### 3.2. Characterization of PSiPs

The internal microstructure of PSiPs was studied using TEM before functionalization. Figure 5a,b shows the internal morphology of the PSiPs as prepared, which showed, due to the modified synthesis parameters, higher particle size and dispersion (1.5 ± 0.5 μm) with respect to previously synthesized particles [25]. These are intrinsically anisotropic with well-defined columnar pores, which present an open structure as determined from the top view image (Figure 5b), in which electron beam and PSiPs pores presented identical directions. The images from single pores (Figure 5c) established a difference between the wall edge and the corrugated internal pore. Higher magnification images (see Figure 5d) allowed the identification of Si nanocrystals in the internal pore wall by slightly tilting the sample until Si(100) planes were resolved.

The SEM images shown in the bottom part of Figure 5 highlight the surface morphology of the functionalized PSiPs in comparison with the morphology of the original PSiPs. Larger particles were chosen to highlight the homogeneity of the surface termination. The latter shows a spongy surface with an average pore aperture of 40 nm. Wide fields were selected to identify the homogeneity of the surface at the scales where the pores start to be observable, while insets show a higher magnification view of the pore structure and distribution.

We note the similarity between the pristine PSiPs (Figure 5e) and the functionalized structures (Figure 5f,g). In particular, there is no film or colloidal structure associated with the organosilane modification obstructing the PSiPs pores. These results support the view that the modification takes place through a mild assembly of the organosilane. In other words, the assembly is heterogeneous and does not consist of adsorbed structures stemming from a previous (in liquid) homogeneous nucleation. In conclusion, this morphological study confirms that the activated assembly process gives rise to a structure with no remarkable topographic addition.

It is widely accepted that as prepared PSiPs exhibit Si–H bonds on the surface [37]. The FTIR spectrum of freshly-etched PSiPs (Figure 6), reflects the presence of these species. We can clearly distinguish the scissoring modes at 624, 662 and 901 cm^−1^, and on the other hand, the stretching modes at 2085 and 2108 cm^−1^ [38]. After VLAO process, the peaks corresponding to SiH_x_ modes tend to disappear, confirming the chemical modification of the pore surface. We confirmed the mildly activated oxidation of PSiPs through bands at 940–977 cm^−1^ (δ(-O_y_Si-H_x_)) and 2195–2249 cm^−1^ (ν(-O_y_Si-H_x_)). Additionally, we observed in APTS and PFDS silanized samples, the presence of new peaks due to the aliphatic carbon chains at 575, 1318, 1382, 1472, 2849 and 2917 cm^−1^ corresponding to surface integrated CH_2_–C bonds.

Oxidation and aliphatic presence are especially evident in APTS-PSiPs. The peaks at 780, 1088, 1627 and 3435 cm^−1^ can be attributed to N–H_2_, C–N, δ(N–H) and ν_AS+S_ (NH_2_), respectively. However, in the case of the two latter modes, their identification is compromised due to an overlapping with the broad water absorption bands caused by the moisture absorbed on the KBr disk. However, in the case of the band at 3435 cm^−1^ we can see that it increases once the PSiPs surface has been functionalized with APTS. This is due to water adsorption, which is consistent with an increasingly polar character induced by the amino groups.

In the spectra corresponding to samples with PFDS, the SiH_x_ modes partially remain after VLAO. Additionally, the CF_3_ group presents a sharp characteristic band at the low wavenumber edge (858 cm^−1^). The CF_2_ group, owing to its four characteristic bands at 1133, 1149, 1208 and 1238 cm^−1^, also appears to be an ideal group for tracing the functionalization [39]. Overall, the FTIR results demonstrate that oxidation processes of PSiPs take place during VLAO. Furthermore, an organosilane assembly is induced on the PSiPs surface.

The overlapping of some characteristic bands in the FTIR spectra of PSiPs makes the univocal identification of molecular groups difficult. In order to overcome this issue, we used SS-NMR spectroscopy, taking advantage of its chemical sensitivity. Figure 7a shows the ^29^Si NMR spectra of the PSiPs with different organosilanes compared with freshly formed PSiPs. In the spectra of the functionalized PSiPs, the chemical shift of the dominant signal is centered at −96 ppm, with a line width (full width at half maximum, FWHM) of 20 ppm, which can be assigned to the SiH or SiH_2_ structural elements [37]. No signal corresponding to amorphous silicon (*a*-Si) (a very broad signal centered at −40 ppm) was observed.

After functionalization, several peaks appeared, indicating the presence of different environments, which complement the signal of the underlying PSiPs. Relevantly and with respect to the effect of the VLAO on PSiPs oxidation, alterations in Q^n^ contributions were identified—the peaks at −85, −101 and −109 ppm corresponding to germinal hydroxyl silanol sites [(O)_2_Si(OH)_2_, Q^2^], hydroxyl containing silicon sites [(O)_3_SiOH, Q^3^] and cross-linked Si[(O)_4_Si, Q^4^], respectively [40]. The high intensity of these peaks indicates that a significant part of the PSiPs surface is oxidized/hydrolyzed to form Q^n^ structures. In the case of PSiPs functionalized with PFDS, these signals appear as shoulders of a main large peak at −95 ppm corresponding to the unreacted PSi. This means that the fluorosilane is effectively protecting the Si surface against hydration and hydroxylation.

The spectra provide further details of the presence of Si–C bonds and exhibit low intensity T^n^ peaks at −50, −63 and −67 ppm assigned to the Si atoms covalently bonded to organic groups R (T^1^ [(SiO)Si(OEt)_2_(R) and R = –CH_2_CH_2_CH_2_NH_2_ and –CH_2_CH_2_(CF_2_)_7_CF_3_ for APTS and PFDS respectively], T^2^ [(SiO)_2_Si(OEt)(R) and identical meaning for R] and T^3^ [(SiO)_3_Si(R), idem for R]). These peaks confirm the reaction of PSiPs with the two silanes.

As the electronic shielding of the central Si increases, the chemical shift becomes increasingly negative with each additional Si–O–Si linkage [41,42]. It is remarkable that the dominant intensity is observed at T1, which would indicate that the organosilanes are not fully condensed, especially in the case of PFDS (most intense T1 signal). This type of information on the reactivity at the surface is extremely relevant and cannot be extracted by using traditional spectroscopic techniques used for surface characterization of organosilanes such as XPS.

Figure 7b presents the ^13^C spectra of the organosilane functionalized PSiPs. In the spectra corresponding to APTS-PSiPs and PFDS-PSiPs, we observed peaks at 32 and 58 ppm due to the CH_3_ and CH_2_ from unreacted ethoxy groups (OCH_2_CH_3_) [43]. Moreover, in the case of the APTS-PSiPs, the expected three characteristic signals of the aminosilane are present at 8.3 ppm (Si–CH_2_), 17 ppm (CH_2_–CH_2_–NH_2_) and 42.2 ppm (CH_2_–NH_2_). The unfolding and the widening of the latter two signals, as well as the shoulder of the CH_3_ peak at 32 ppm, indicate the presence of a small portion of protonated amine. On the other hand, the signal at 21 ppm corresponds to the intermediate CH_2_ of the aliphatic chain of amine [44].

In the spectrum corresponding to the PFDS-PSiPs, we observed small peaks at 16 ppm (Si–CH_2_–) and 24 ppm (Si–CH_2_–CH_2_–) corresponding to the aliphatic chain. They are both shifted down-field with respect to the equivalent carbons on the aminopropyl chain because of the adjacent CF_2_ groups. The small intensity of the signals confirms the data obtained in the ^29^Si spectrum, indicating that the organosilane condensation is not complete.

Because ^19^F is one of the most useful NMR active nuclei, the ^19^F NMR spectrum was obtained from the PFDS-PSiPs. Indeed, the ^19^F NMR spectra of PSiPs and PFDS-PSiPs in Figure 7c show no similarities. The spectrum corresponding to the PSiPs reference presents two peaks at −93 and −147 ppm corresponding to SiF_3_H and SiF_2_H species, respectively. In the spectrum of PFDS-PSiPs, we can clearly identify the peak corresponding to the CF_3_ group at 83.8 ppm. The CF_2_ groups give rise to peaks at 124.1 and 145 ppm [27].

In addition to the structural information provided by SS-NMR, quantitative studies can be carried out owing to the direct proportionality between the signal intensity and the number of contributing nuclei. In the case of the functionalization with PFDS, we obtained a relation between the peaks corresponding to the CF_2_ and CF_3_ bonds in agreement with the molecular structure of the monolayer precursor (~7).

Combining the results obtained from FTIR spectra with those from NMR spectra, the overall information is consistent with the formation of the organosilane functionalized PSiPs. It appears however, that the ratio of organosilane molecules with three siloxane bonds saturated to the PSiPs surface is low.

### 3.3. Reaction Mechanism

The results from the characterization of PSiFs and PSiPs subject to the VLAO process have shown evidence of both PSi surface and organosilane transformation. Relevantly, the different analytical techniques reinforced the idea that an activation of a surface oxidation of PSi is induced and an instability in hydrolysable siloxane bonds of the organosilane is activated, which favors the formation of Si–O–Si bonds. Figure 8 shows two proposed parallel reactions capable of explaining the observed changes on the PSi surfaces. From the side of the fresh PSi surface, contrary to ultra-violet (UV) light, visible light at room temperature does not provide photons with sufficient energy to promote the homolytic cleavage of the Si–H bonds. However, excitons (electron-hole pairs) are generated during the illumination of PSi at the H–Si surface due to photo-excitation [45]. This facilitates the nucleophilic attack of trace water found in absolute ethanol, leading to a transformation of the original Si–H to a Si–OH bond (Figure 8a). Thus, a hydroxylated PSi surface is formed avoiding the thermal [46] or chemical [16] oxidation processes previously proposed. From the side of the organosilane, it is widely accepted that their tendency towards hydrolysis and condensation of alkoxysilanes, such as APTS and PFDS, is lower than that of transition metal alkoxydes. However, it is known that surface hydroxyl groups are the driving force for the heterogeneous condensation of these alkoxydes (Figure 8b). It is likely that during the progress of the heterogeneous condensation, the dangling bonds from adsorbed APTS or PDFS bind a neighboring molecule, which would be illustrated by the molecular structures identified on PSiFs and PSiPs compatible with slightly oligomerized organosilanes.

Because the proposed VLAO conditions imply a simultaneous exposure of the PSi structures and organosilanes to visible light, alternative reactions could be activated at the organosilanes. At this point, it is relevant to maintain the low concentration of the organosilanes (0.2%), which ensures that the light absorbed, is well below the light absorbed in PSi. Under these conditions, the rates of production of homogeneous silane–silane reactions (whatever their nature can be) must remain well below the production of heterogeneous PSi-silane reactions. This gives further support to the proposed chemical route for the VLAO process on PSi structures.

## 4. Conclusions

A condensation process has been applied to the synthesis of organosilane functionalized PSiFs and PSiPs using APTS and PFDS at low concentrations of 0.2%. Visible light activation was shown to promote PSi surface oxidation making the reaction with organosilanes viable. The process leads to homogeneous surfaces with no traces of silane derived colloidal structures. This suggests that organosilane coverage is below one monolayer and makes the process useful as a model for the analysis of organosilane-PSi interfaces. In particular, SS-NMR and FTIR proved to be powerful techniques to obtain spectroscopic features regarding the organosilane assemblies on PSiPs. Analogue information is obtained by using XPS and ToF-SIMS on PSiFs. Indeed, while the FTIR and XPS analyses strongly supported organic surface bound moieties resulting from the functionalization reactions, SS-NMR and ToF-SIMS provided more detailed information from the interface species, in particular, the surface oxidation induced during VLAO process and the presence of slightly oligomerized organosilane molecules. Using ^13^C SS-NMR, we differentiated aliphatic carbons within the same chain (i.e., as in the perfluorodecyl group) and with respect to the other silanes (i.e., the perfluorodecyl, with respect to the aminopropyl). This is especially relevant as there is no surface spectroscopic technique with the requisite chemical sensitivity to reveal these features.

The combined use of ^29^Si and ^19^F SS-NMR was found to be extremely useful in the determination of silane binding efficiency. PFDS showed a lower degree of condensation with PSiPs (dominant T1 in ^29^Si spectra), suggesting that it provides the highest hydroxylation protection (reduced formation of Q^n^ structures). This difference has its origin in the influence of the non-hydrolysable radical of the organosilane in the distribution of charge in the molecule. This is known to slightly influence the susceptibility of the alkoxy groups to homogeneous and heterogeneous condensation. In fact, the hydrophobic nature of the perfluorodecyl group may have a direct impact in the retardation of formation of the condensation by-product molecules, constituting a steric hindrance effect.

As a prospective remark, for the sake of functionalization efficiency, a study of the optimization of organosilane concentration towards increased coverage should be considered. Since PSi can potentially be used to allocate nanomaterials and biomolecules to provide additional functionalities, this study highlights the interest on the multi-technique surface analysis of any nano or biomolecular complexes conjugated with PSi.

## Figures and Tables

**Figure 1 materials-12-00131-f001:**
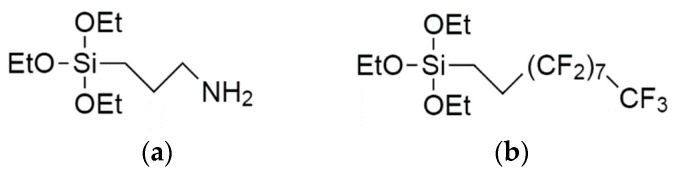
Molecular structure of (**a**) 3-aminopropyl-triethoxy-silane (APTS) and (**b**) 1*H*,1*H*,2*H*,2*H*-perfluorodecyl-triethoxy-silane (PFDS).

**Figure 2 materials-12-00131-f002:**
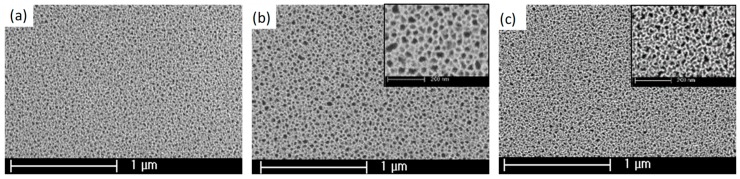
Field emission scanning electron microscopy (FESEM) images of the surface of PSiFs before (**a**) and after assembly of APTS (**b**) and PFDS (**c**).

**Figure 3 materials-12-00131-f003:**
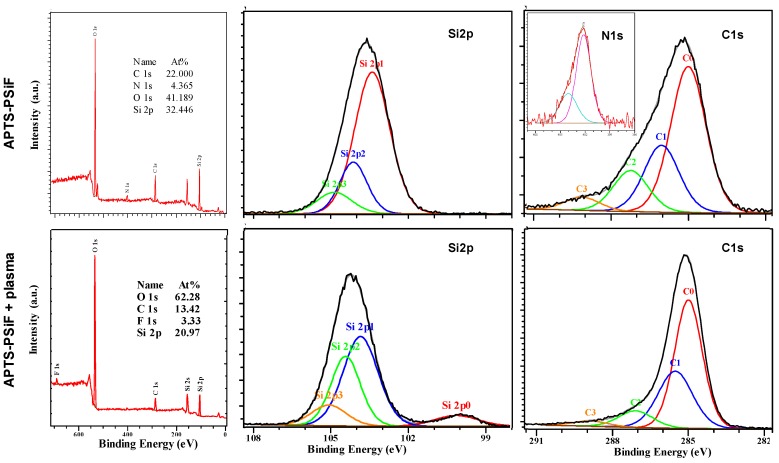
X-ray photoelectron spectroscopy (XPS) analysis of an APTS functionalized PSiF before (**top**) and after (**bottom**) an Ar plasma etching process. From left to right: widescan, Si2p and C1s spectra. Note: N1s inset to the fresh APTS-PSiF sample.

**Figure 4 materials-12-00131-f004:**
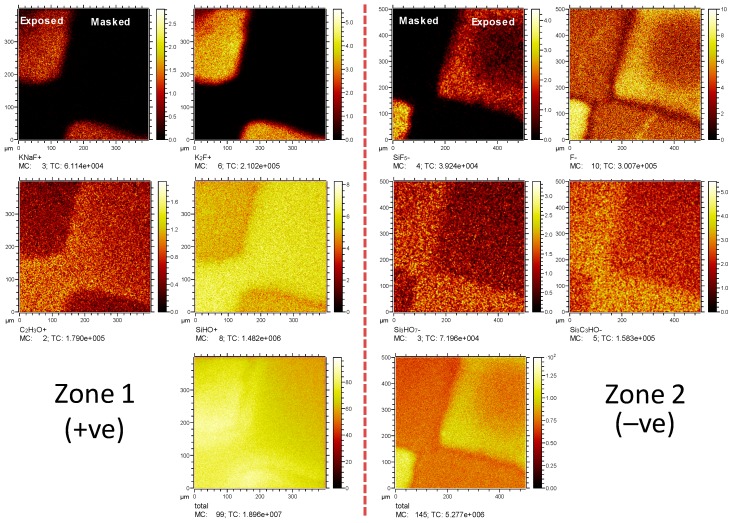
Time of flight secondary ion mass spectroscopy (ToF-SIMS) chemical maps from a micropatterned PSiF surface after PFDS functionalization. **Left**: KNaF^+^, K_2_F^+^, C_2_H_2_O^+^, SiOH^+^ and total positive ions (from left to right and top to bottom). **Right**: SiF_5_^−^, F^−^, Si_3_HO_7_^−^, Si_3_C_3_HO^−^ and total negative ions (from left to right and top to bottom).

**Figure 5 materials-12-00131-f005:**
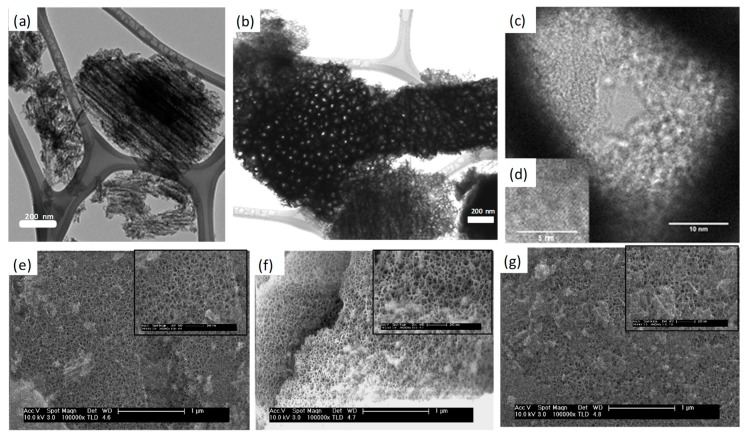
Transmission electron microscopy (TEM) lateral (**a**) and top (**b**) images of columnar PSiPs and (**c**) TEM image of a pore and (**d**) detail of a Si nanocrystal revealing (100) planes. FESEM images of (**e**) the surface of PSiPs, and PSiPs functionalized with (**f**) APTS and (**g**) PFDS.

**Figure 6 materials-12-00131-f006:**
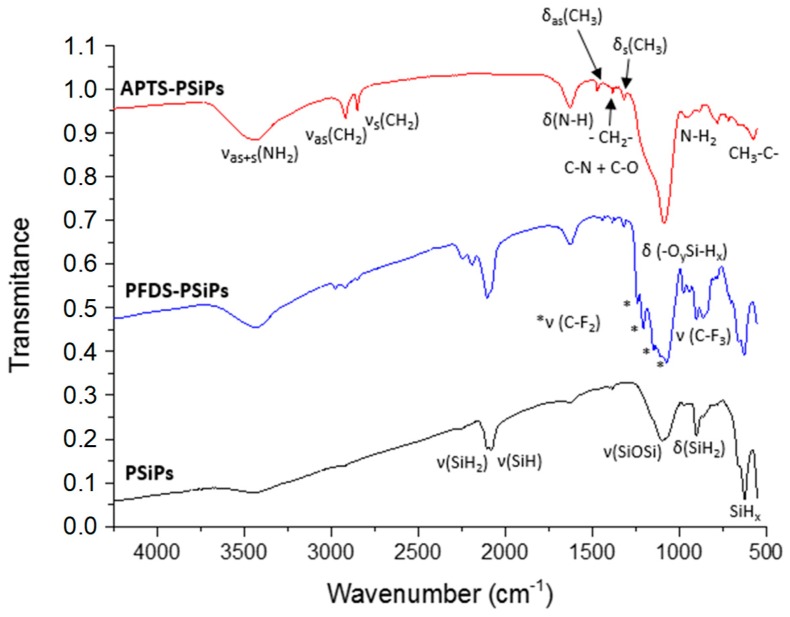
Fourier transformed infrared spectroscopy (FTIR) spectra of APTS and PFDS functionalized PSiPs compared with the spectrum corresponding to freshly formed PSiPs.

**Figure 7 materials-12-00131-f007:**
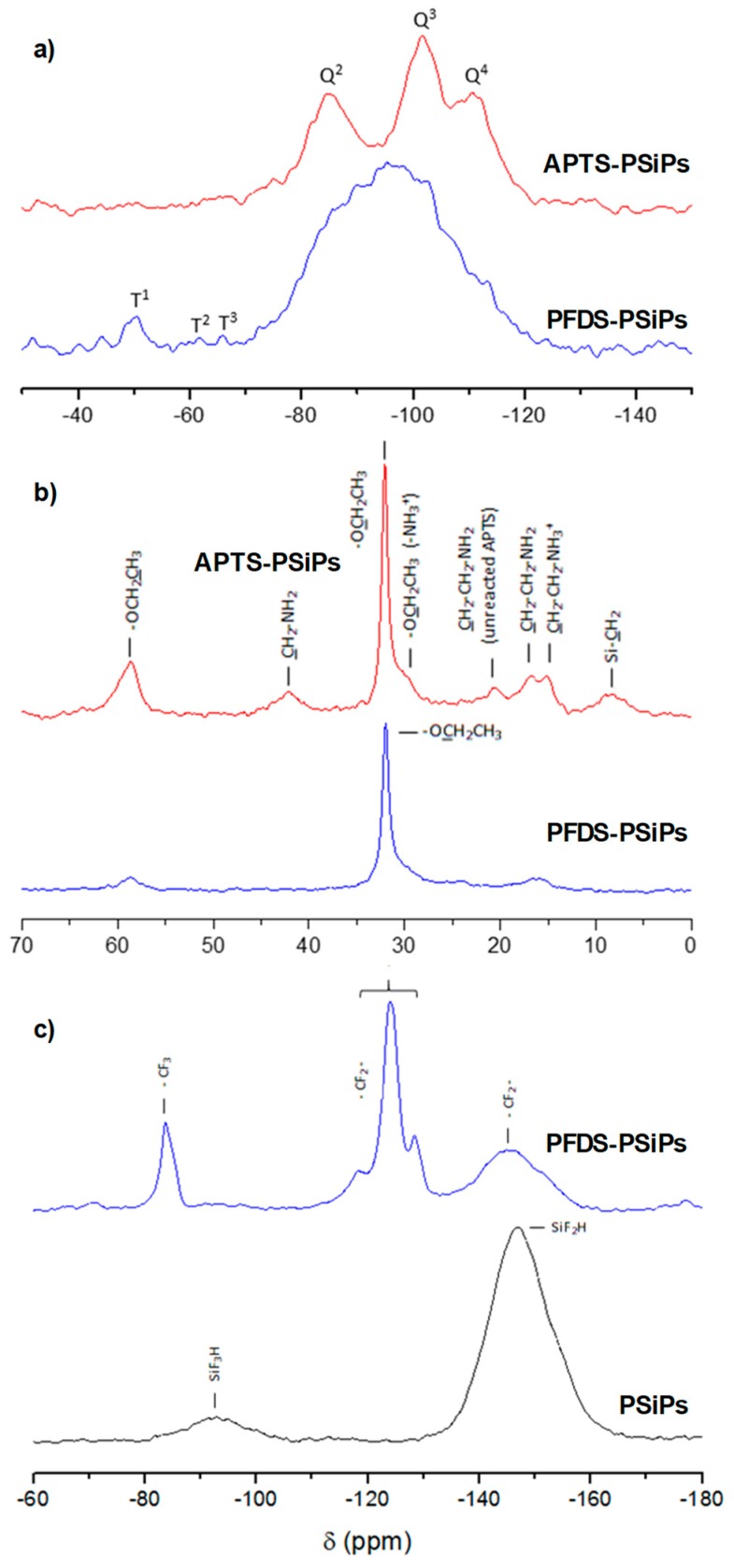
^29^Si (**a**) and ^13^C (**b**) nuclear magnetic resonance (NMR) spectra of the APTS and PFDS functionalized PSiPs. ^19^F NMR spectra of PFDS PSiPs and fresh PSiPs (**c**).

**Figure 8 materials-12-00131-f008:**
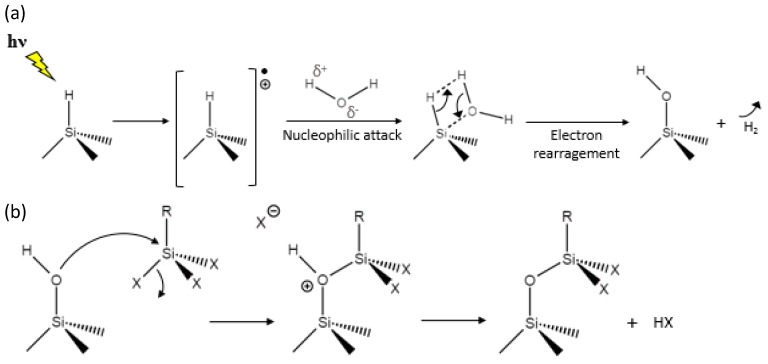
Proposed mechanism for the (**a**) visible light activation of PSi and (**b**) further condensation of the organosilanes on the hydroxylated PSi surface. X = OEt. R = –CH_2_CH_2_(CF_2_)_7_CF_3_ and –CH_2_CH_2_CH_2_NH_2_ for PFDS and APTS, respectively.

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
