# Peer review of "Visible Light Assisted Organosilane Assembly on Mesoporous Silicon Films and Particles"

_materials, 2019, doi:10.3390/ma12010131_

Round 1

Reviewer 1 Report

The work is well presented. The strategies for the functionalization of the films and the nanoparticles are well stablished by the authors and the characterisation is correctly developed.

Particularly, I liked very much the use they do of Solid State NMR (Si-29, C-13 and F-19) because of the high quality of the structural information obtained.

The conclusions of the paper are well formulated.

Author Response

Dear Reviewer,

Thank you for the positive impressions about our document. We also agree that the use of SS-NMR is a strong point of the work. We are prone to generalizing these analyses in forthcoming  organosilane conjugation studies.

Thank you again

Reviewer 2 Report

In this paper, the authors reported using visible light assisted oxidation of Si-H on porous silicon. While the reviewer finds some of the underlying concepts relatively interesting, there are several major issues that needs to be address prior to acceptance.

1.     The reviewer felt that the major attraction of this work should be the direct light-assisted oxidation of Si-H bond on porous silicon but this point was not adequately addressed in the abstract. Hence in certain sense, the point has not been driven at the first glimpse and should be given a greater weightage in the abstract.

2.     On the basis of the two silanes used in the work, shouldn’t contact angle measurement be a suitable means of determinating wettability on intact porous silicon film without sonication?  Such data can help to validate the success in silanization?

3.     “Although the porosity remains within the same order of magnitude for both samples, the resulting mean pore size is slightly bigger for APTS functionalization (roughly 40 nm) as compared for the PFDS functionalization (circa 25 nm). The origin of this difference may reside in different modes of adsorption of the organosilanes due to the characteristic length and interaction modes of the aminopropyl and perfluorodecyl groups”.  The reviewer does not fully understand the meaning of this sentence.  What differences in mode of adsorption had occurred for the perfluorodecyl and aminopropyl silane? The pore sizing of the porous silicon was a mechanic process from electrochemical anodization and the subsequent silanization chemistry may have weakened the silicon back bonding but considering that both silanes should effectively graft at similar silanol ends, shouldn’t the aftermath chemistry be relatively identical and not resulting in any of the difference structural changes observed?  The authors may want to refer on the following works on Si-O linkage and the subsequent back bond weakening (J. Chem. Soc., Perkin Trans. 2, 2002, 23–34 and Scientific Reports, vol. 5, no. 11299 (2015)) There should be a need to further elaborate on this matter.

4.     XPS spectrum is inadequate in the revelation of the chemistry. Firstly, the authors should include the XPS of freshly made porous silicon as a comparison.  Secondly, It may be necessary to include N1s and F1s to give a fuller picture.  Finally, there is a lack of discussion on the assignments, especially the C1s for the perfluorodecyl as the reviewer is hard-press to identify the nominal CF peaks above 290 eV.

5.     From XPS and FTIR data, it suggested that the grafting of perfluorodecyl was lower than that of the aminopropyl.  Perhaps the authors may like to explain why this was so? Whether if steric hindrance may play a role in this matter will remain as an interesting proposition

6.     How long was the illumination done for? How long was the silanization performed? This was not entirely clear in the experimental section and perhaps the authors should improve on this.

7.     The notion of exciton driven reactivity towards water is interesting but in the presence of silane, the reaction may also occur at the NH2 end other in the form of nucleophilic addition.  Due to the manner that the experimental protocol was written, the reviewer is not clear whether this illumination was done in conjugation with the addition of silanes or not. Hence the experimental section needs to be written in a more concise manner.

Author Response

Dear Reviewer,

Thank you for the critical reading of our article. We can appreciate that your comments will lead to an improvement of our contribution. Please, enclosed the replies to the criticisms you raised.

1.     The reviewer felt that the major attraction of this work should be the direct light-assisted oxidation of Si-H bond on porous silicon but this point was not adequately addressed in the abstract. Hence in certain sense, the point has not been driven at the first glimpse and should be given a greater weightage in the abstract.

We agree that the oxidation of the surface is something that is key in the mechanism of organisilanization. We have thus modified the document according to this suggestion. 

2.     On the basis of the two silanes used in the work, shouldn’t contact angle measurement be a suitable means of determinating wettability on intact porous silicon film without sonication?  Such data can help to validate the success in silanization?

We agree again with the observation of the referee. We think however that the behaviour of the contact angle (that we have studied) is part of a functional evaluation of the organisilanized PSi structures. We anticipate that water contact angles of 117º and 36 +/- 4º degrees are obtained for PFDS and APTS functionalization, respectively. These data are being the subject of continuous work detailing the permeability of the functionanlized structures and their biomedical applications. We understand the current work as a physicochemical evidence of the photo-assisted process. 

3.     “Although the porosity remains within the same order of magnitude for both samples, the resulting mean pore size is slightly bigger for APTS functionalization (roughly 40 nm) as compared for the PFDS functionalization (circa 25 nm). The origin of this difference may reside in different modes of adsorption of the organosilanes due to the characteristic length and interaction modes of the aminopropyl and perfluorodecyl groups”.  The reviewer does not fully understand the meaning of this sentence.  What differences in mode of adsorption had occurred for the perfluorodecyl and aminopropyl silane? The pore sizing of the porous silicon was a mechanic process from electrochemical anodization and the subsequent silanization chemistry may have weakened the silicon back bonding but considering that both silanes should effectively graft at similar silanol ends, shouldn’t the aftermath chemistry be relatively identical and not resulting in any of the difference structural changes observed?  The authors may want to refer on the following works on Si-O linkage and the subsequent back bond weakening (J. Chem. Soc., Perkin Trans. 2, 2002, 23–34 and Scientific Reports, vol. 5, no. 11299 (2015)) There should be a need to further elaborate on this matter.

The reviewer is again right in this judgement. We have modified our comments and attributed the differences to the slight heterogenous pore size obtained in single preparations due to radial difference of the electric field on the surface of the Si cell and to batch to batch production, since the electrolyte losses little by little its chemical potential as the samples in one series are produced. We have taken good care of proposed bibliography. We consider however that the proposed articles take the discussion far from the experimetnal conditions due to: a) they concern flat Si instead of PSi, b) They imply the use of UV instead of visible light. In this sense, we have referenced in previous works the two following works (J. Am. Chem. Soc. 127 (2005) 2514e2523, Jpn. J. Appl. Phys. 47 (2008) 5659e5664), which go very much in line with the proposed bibliography. 

4.     XPS spectrum is inadequate in the revelation of the chemistry. Firstly, the authors should include the XPS of freshly made porous silicon as a comparison.  Secondly, It may be necessary to include N1s and F1s to give a fuller picture.  Finally, there is a lack of discussion on the assignments, especially the C1s for the perfluorodecyl as the reviewer is hard-press to identify the nominal CF peaks above 290 eV.

The reviewer is somehow right, but from our point of view, it is safer and more advanced to find the characteristics of the plasma etched PSiFs. The comparison with the chemical state of freshly  formed PSi is straight forward from previous (external and own) bibliography. The most relevant aspect is the re-emergence of a Si-Si component in the Si2p core level of the plasma etched PSiF, which is dominant for a typical fresh PSi sample (Appl Phys Lett 2007;91:103113, J Biomed Mater Res Part A 2012:100A:1615–1622).

The most important N1s contribution has been included (for the APTS functionalized surface), but the signal after plasma etching is noisy. At the same time, the F2p peak is noisy also and we recognize that it was not even scanned as core level. The reviewer must take into account that there is no XPS analysis for PFDS functionalization since the perfluorodecyl group is highly labile under X-ray irradiation and the products specially damaging for the spectrometer components (source of contamination and electronics damage). That is one reason why more local tof-SIMS was used for this sample. There is thus no contribution over 290 in the C1s spectra, 

5.     From XPS and FTIR data, it suggested that the grafting of perfluorodecyl was lower than that of the aminopropyl.  Perhaps the authors may like to explain why this was so? Whether if steric hindrance may play a role in this matter will remain as an interesting proposition

 The reviewer is right. The different non hydrolyzable chain of analogue alkoxysilanes modify slightly the hydrolysis and condensation kinetics of the hydrolyzable groups. In our case, the hydrophobic properties of the perfluorodecyl group slow down the rate of production of these reactions, and can be thus considered as inducer of a steric hindrance. See new paragraph in the conclusions section. 

6.     How long was the illumination done for? How long was the silanization performed? This was not entirely clear in the experimental section and perhaps the authors should improve on this. 

 We have clarified in the present version that illumination is simultaneous to the silanization process. The time was 10 min for APTS and 30 min for PFDS, due in fact to the lower rates of assistance of the process to the PFDS molecule as discussed in the previous point. The kinetics of the reaction has been studied for another silane (GPTMS, see ref [18]). In this case we just looked for conditions that provided an equivalent Si-O FTIR band evolution for APTS and PFDS.  

7.     The notion of exciton driven reactivity towards water is interesting but in the presence of silane, the reaction may also occur at the NH2 end other in the form of nucleophilic addition.  Due to the manner that the experimental protocol was written, the reviewer is not clear whether this illumination was done in conjugation with the addition of silanes or not. Hence the experimental section needs to be written in a more concise manner.

We have clarified this relevant experimental point. In fact, PSi structures and organosilanes are exposed simultaneously to the visible light. We note at this point that the light absorbed by the solution is minima (note organosilane concentration of 0.2%), while it is enormous at the PSi surface. We consider thus that, even if possible, the rate of production of solution products must be orders of magnitude below the production of interface products. A comment on reference to this has been included in the "reaction mechanism" sub-section. 

Round 2

Reviewer 2 Report

The manuscript had been improved on the basis of previous comments and should be deemed suitable for publication.